# Modeling of Red Blood Cells in Capillary Flow Using Fluid–Structure Interaction and Gas Diffusion

**DOI:** 10.3390/cells11243987

**Published:** 2022-12-09

**Authors:** Ling An, Fenglong Ji, Yueming Yin, Yi Liu, Chichun Zhou

**Affiliations:** 1School of Engineering, Dali University, Dali 671003, China; 2School of Textile Materials and Engineering, Wuyi University, Jiangmen 529020, China

**Keywords:** diffusion capacity, cell deformation, cell clustering, capacity flow

## Abstract

Red blood cell (RBC) distribution, RBC shape, and flow rate have all been shown to have an effect on the pulmonary diffusing capacity. Through this study, a gas diffusion model and the immersed finite element method were used to simulate the gas diffusion into deformable RBCs running in capillaries. It has been discovered that when RBCs are deformed, the CO flux across the membrane becomes nonuniform, resulting in a reduced capacity for diffusion. Additionally, when compared to RBCs that were dispersed evenly, our simulation showed that clustered RBCs had a significantly lower diffusion capability.

## 1. Introduction

Through the transport of blood, multicellular organisms’ cells receive nutrients, oxygen, and waste removal from the blood circulatory systems of humans. Contrary to traditional opinion, blood is actually a complex suspension of deformable cells, proteins, platelets, and plasma, as opposed to the homogeneous fluid that is typically thought of as blood. The major cause of death in developing countries is currently cardiovascular disease. Numerous arterial and vascular illnesses have poorly understood short and long-term pathways of progression, which has a direct impact on the formulation of therapeutic strategies as well as disease diagnosis and prevention. Thus, knowledge of blood composition, as well as its transportation and rheological properties, has major clinical value.

The impact of blood flow on the transfer of gases (CO_2_, CO, O_2_, etc.) in the alveolar region of the lungs has recently drawn increasing attention. The pulmonary diffusive gas transport is influenced by RBC distribution in alveolar capillaries, particularly the intake of Carbon Monoxide (CO). Diffusing capacity is a measure of the lung’s effectiveness at transferring gas. Recent research has demonstrated that while clustered distribution of RBC results in the lowest outcomes, uniform distribution produces the maximum CO diffusing capacity. The pulmonary diffusing capacity, in particular CO absorption, is impacted by differences in RBC shape [1]. According to research, uniform capillaries with diameters that are similar to those of RBCs have been shown to cause significant RBC deformation, which lowers CO absorption and can be further amplified by RBC clustering during capillary contraction [2]. Errors are introduced into morphometric estimates of diffusing capacity by both the distribution and distortion of RBC [3]. The investigation into the effects of these two factors is only at the static and simple geometry stages (for both capillary and RBC). The gases oxygen and carbon dioxide inside the pulmonary capillaries were numerically analyzed by Whiteley et al. [4] They observed that the amount of carbon dioxide that was discharged from blood into alveolar gas relies on oxygen concentration and that there is barely any restriction on oxygen transport when oxygen concentrations are normal or high. A two-dimensional finite element model was developed by Frank et al. [5] to investigate oxygen diffusion in pulmonary capillaries. We employed a deformed RBC shape that was digitally captured from an image and fitted with a cubic spline. Such processes have not been the subject of hydrodynamic modeling. Our simulation may assist in compensating for such inaccuracy and improving the interpretation of experimental measurement by dynamically assessing these two effects using real geometries. And the carbon monoxide diffusing capacity (DLCO) test has developed into one of the most effective standard lung function tests used in laboratories today [6], so we chose CO as our simulation study object. Although we focused on the particular instance of CO diffusion, the modeling methods and analysis employed here can be simply applied to O_2_ or CO_2_.

In this study, we combined fluid flow with gas diffusion to study the fluid–structure interaction between RBCs. With the aid of the immersed finite element method (IFEM), the RBCs flow in capillary is resolved. In a capillary flow, RBC motion and deformation are linked to a gas diffusion issue. Because the small domain size and number of RBCs used in the simulation, as well as other intricate biochemical elements present in real blood that are not taken into account by our cell-cell interaction model, make it difficult to make precise comparisons to the experimental results. This issue will be addressed in our future works. However, to our knowledge, this is the first research that links the microscopic mechanism of RBC aggregation to the macroscopic blood viscosity via direct 3D numerical simulation.

The following is a summary of the paper. In Section 2, we first go over the IFEM formulas and method. Section 3 provides the numerical models for RBCs and gas diffusion. Section 4 studies the CO diffusion capability during the RBC deformation process and under various distributions. We summarize our findings and recommendations for further study in Section 5.

## 2. Immersed Finite Element Method Review 

### 2.1. Formulations of IFEM

The IFEM was created by Zhang et al. [7] to address the issue of fluid–deformable structure interaction, and Liu et al. [8,9,10,11] and Kim et al. [12] have used it to simulate issues with particle flow, nanowire assembly, and red blood cell aggregation. The IFEM formulas are summarized in the section that follows.

Let us look at an incompressible, three-dimensionally deformable structure in $\Omega^{s}$ that is entirely submerged in an incompressible fluid domain $\Omega^{f}$. The solid and the fluid form a domain Ω together, but they do not intersect:(1)$$\Omega^{f}\;\cup^{\text{​}}\Omega^{s}=\Omega ,$$
(2)$$\Omega^{f}\;\cap^{\text{​}}\Omega^{s}=\emptyset .$$

The solid domain can occupy a finite volume in the fluid domain, in contrast to the IB formulation. The union of two domains can be thought of as one incompressible continuum with a continuous velocity field since we assume that both fluid and solid are incompressible. A Eulerian fluid mesh was used in the computation since the fluid covers the entire domain; a Lagrangian solid mesh was then built on top of the Eulerian fluid mesh. When creating the momentum and continuity equations, some factors related to the coexistence of fluid and solid in $\Omega^{s}$ must be taken into account.

The time-invariant position vector x represents the fluid grid in the computational fluid domain Ω, while X^s^ and x^s^ (X^s^, t), respectively, represent the material points of the structure in the initial solid domain $\Omega^{s}$ and the current solid domain $\Omega^{s}$. In the solid variables, the superscript s was used to distinguish between the solid and fluid domains.

In the fluid calculations, the velocity **v** and the pressure *p* are the unknown fluid field variables; the solid domain requires the computation of the nodal displacement u**^s^**, which is defined as the difference between the current and the beginning coordinates: u^s^ = x^s^ − X^s^. The velocity **v^s^** is the material derivative of the displacement du**^s^**/dt. Since displacement and acceleration are not explicitly utilized in fluid calculations, they are not described under the column for the fluid domain. Table 1 provides a summary of the definitions of the spatial coordinate, displacement, velocity, and acceleration in both the fluid and solid domains.

The derivative of the Cauchy stress *σ* and the external force applied to the continuum balance the inertial force of a particle in a continuum.
(3)$$\rho \frac{dv_{i}}{dt}=\sigma_{ij,j}+f_{i}^{ext}.$$

We can split the inertial force into two components within the solid domain, $\Omega^{s}$, if the solid density $\rho^{s}$ differs from the fluid density $\rho^{f}$, i.e., $\rho =\rho^{s}$ in $\Omega^{s}$ and $\rho =\rho^{f}$ in $\Omega^{f}$:(4)$$\rho \frac{dv_{i}}{dt}=\{\begin{array}{c} \rho^{f}\frac{dv_{i}}{dt},\;x\in \Omega /\Omega^{s},\;\\ \rho^{f}\frac{dv_{i}}{dt}+\left(\rho^{s}-\rho^{f}\right)\frac{dv_{i}}{dt},\;x\in \Omega^{s}. \end{array}$$

Note that we describe the fluid domain $\Omega^{f}$ as the total domain minus the solid domain $\Omega^{s}$, namely $\Omega /\Omega^{s}$, to highlight the point of addition and removal of sub-domains. Additionally, the gravitational force is used to explain the external force $f_{i}^{ext}$ in this thesis, and as a result, we are able to
(5)$$f_{i}^{ext}=\{\begin{array}{c} \rho^{f}g_{i},\;x\in \Omega /\Omega^{s},\;\\ \rho^{f}g_{i}+\left(\rho^{s}-\rho^{f}\right)g_{i},\;x\in \Omega^{s}. \end{array}$$

Since the entire domain Ω is the computational fluid domain, we disregard the hydrostatic pressure and rewrite Equation (5) as
(6)$$f_{i}^{ext}=\{\begin{array}{c} 0,\;x\in \Omega /\Omega^{s},\;\\ \left(\rho^{s}-\rho^{f}\right)g_{i},\;x\in \Omega^{s}. \end{array}$$

Additionally, the derivative of the Cauchy stress in Equation (3) can be divided into the following parts:(7)$$\sigma_{ij,j}=\{\begin{array}{c} \sigma_{ij,j}^{f},\;x\in \Omega /\Omega^{s},\;\\ \sigma_{ij,j}^{f}+\sigma_{ij,j}^{s}-\sigma_{ij,j}^{f},\;x\in \Omega^{s}.\; \end{array}$$

The fluid stress in the solid domain of Equation (7) is generally substantially smaller than the corresponding solid stress [13], as can be seen. Assuming that FSI stands for fluid–structure interaction, let us define the force acting within the domain $\Omega^{s}$ as $f_{i}^{FSI,s}$:(8)$$f_{i}^{F\mathrm{SI},s}=-(\rho^{s}-\rho^{f})\frac{dv_{i}}{dt}+\sigma_{ij,j}^{s}-\sigma_{ij,j}^{f}+\left(\rho^{s}-\rho^{f}\right)g_{i},\;x\in \Omega^{s}.$$

Naturally, the Lagrangian description is used to determine the interaction force $f_{i}^{FSI,s}$ in Equation (8). Additionally, the interaction force between the solid domain and the computational fluid domain is distributed using a Dirac delta function δ:(9)$$f_{i}^{F\mathrm{SI},s}\left(x,t\right)=\int_{\Omega^{s}}f_{i}^{F\mathrm{SI},s}\left(X^{s},t\right)\delta (x-x^{s}\left(X^{s},t\right))d\Omega .$$

Thus, by adding the fluid terms and the interaction force, the governing equation for the fluid may be formed as follows:(10)$$\rho^{f}\frac{dv_{i}}{dt}=\sigma_{ij,j}^{f}+f_{i}^{\mathrm{FSI}},\;x\in \Omega .$$

We only need to apply the incompressibility constraint once to the entire domain because we believe the entire domain to be incompressible:(11)$$v_{i,i}=0.$$

We added various velocity field variables $v_{i}^{s}$ and $v_{i}$ to describe the motions of the solid in the domain $\Omega^{s}$ and the fluid within the entire domain Ω in order to distinguish between the Lagrangian description for the solid and the Eulerian description for the fluid. The Dirac delta function was used to couple the two velocity fields:(12)$$v_{i}^{s}\left(X^{s},\;t\right)=\int_{\Omega }v_{i}\left(x,\;t\right)\delta (x-x^{s}\left(X^{s},\;t\right))d\Omega .$$

Assuming that there is no traction imparted to the fluid boundary, or that $\int_{\Gamma_{h_{i}}}\delta v_{i}h_{i}d\Gamma =0$, we can obtain the final weak form (with stabilizing terms) by applying integration by parts and the divergence theorem:(13)$$\begin{array}{l} \int_{\Omega }\left(\delta v_{i}+\tau^{m}v_{k}\delta v_{i,k}+\tau^{c}\delta p,\;i\right)\\ \;\;\;\;\left[\rho^{f}(v_{i,t}+v_{j}v_{i,j})-f_{i}^{\mathrm{FSI}}\right]d\Omega +\underset{\Omega }{\int }\delta v_{i,j}\sigma_{ij}^{f}d\Omega \\ \;\;\;\;-\underset{e}{\sum }\int_{\Omega_{e}}\left(\tau^{m}v_{k}\delta v_{i,k}+\tau^{c}\delta p,\;i\right)d\Omega \\ \;\;\;\;+\int_{\Omega }\left(\delta p+\tau^{c}\delta \tau^{c}v_{i,i}\right)v_{j,j}d\Omega =0. \end{array}$$

The Newton–Raphson method was used to resolve the nonlinear systems. Additionally, we used the GMRES iterative algorithm to increase computing efficiency and compute residuals using matrix-free methods [14,15].

Note that we neglected the fluid stress within the solid domain for the sake of conciseness. The integration domain changed from $\Omega^{s}$ to $\Omega_{0}^{s}$ by the transition of the weak form from the updated Lagrangian to the entire Lagrangian description. The Jacobian determinant in the solid domain is 1, and since we are considering incompressible fluid and solid, the translation of the weak form to the entire Lagrangian description results in the following:(14)$$\int_{\Omega_{0}^{s}}\delta u_{i}\left[\left(\rho^{s}-\rho^{f}\right){\ddot{u}}_{i}^{s}-\frac{\partial P_{ij}}{\partial X_{j}}-\left(\rho^{s}-\rho^{f}\right)g_{i}+f_{i}^{\mathrm{FSI},s}\right]d\Omega_{0}^{s},$$
where the first Piola-Kirchhoff stress $P_{ij}$ is defined as $P_{ij}=JF_{ik}^{-1}\sigma_{kj}^{\;s}$ and the deformation gradient $F_{ij}$ as $F_{ij}=\partial x_{i}/\partial X_{j}$.

We may rewrite Equation (14) using Integration by Parts and the Divergence Theorem:(15)$$\begin{array}{l} \int_{\Omega_{0}^{s}}\delta u_{i}\left(\rho^{s}-\rho^{f}\right){\ddot{u}}_{i}^{s}d\Omega_{0}^{s}+\int_{\Omega_{0}^{s}}\delta u_{i,j}P_{ji}d\Omega_{0}^{s}\\ \;\;\;\;\;\;-\int_{\Omega_{0}^{s}}\delta u_{i}\left(\rho^{s}-\rho^{f}\right)g_{i}d\Omega_{0}^{s}\\ \;\;\;\;\;\;+\int_{\Omega_{0}^{s}}\delta u_{i}f_{i}^{\mathrm{FSI},s}d\Omega_{0}^{s}=0. \end{array}$$

Again, keep in mind that the boundary integral terms on the interface between the fluid and the structure will cancel each other for both the fluid and solid domains and that, for brevity, they are not included in the corresponding weak forms.

The second Piola-Kirchhoff stress $S_{ij}$ and the Green–Lagrangian strain $E_{ij}$ were utilized in the complete Lagrangian formulation for structures with large displacements and deformations:(16)$${\;S}_{ij}=\frac{\partial W}{\partial E_{ij}}{\;\mathrm{and}\;E}_{ij}=\frac{1}{2}\left(C_{ij}-\delta_{ij}\right),$$
where $P_{ij}=S_{ik}F_{ij}$ allows us to derive the first Piola-Kirchhoff stress $P_{ij}$ from the second Piola-Kirchhoff stress.

Finally, the discretized form of Equation (12) in the interpolation from the fluid onto the solid grid can be represented as
(17)$$v_{iI}^{s}=\underset{J}{\sum }v_{iJ}\left(t\right)\phi_{J}\left(X_{J}-x_{I}^{S}\right),\;X_{J}\in \Omega_{\phi I}.$$

By adding the velocities at the fluid nodes within the influence domain $\Omega_{\varphi I},$ it is possible to derive the solid velocity $v_{I}^{s}$ at node I in this case. The dispersion of the solid onto the fluid grid involves a twofold process. The discretized form of Equation (9) is expressed as
(18)$$f_{iJ}^{\mathrm{FSI}}=\underset{I}{\sum }f_{iI}^{\mathrm{FSI}}\left(X^{s},\;t\right)\phi_{I}\left(x_{J}-x_{I}^{S}\right),\;x_{I}^{S}\in \Omega_{\phi J\;}$$

Using the solid particles in Equation (17) to interpolate the fluid velocities onto (16), the fluid was constrained to solid material locations within the solid domain. When the solid mesh is at least two times denser than the surrounding fluid mesh, not only is the no-slip boundary condition on the solid’s surface guaranteed, but the fluid is automatically stopped from penetrating the solid. Based on the numerical evidence, this heuristic criterion requires more research.

### 2.2. Computational Algorithm of IFEM



(19)
$$\;f_{iI}^{FSI,s}=-f_{iI}^{inert}-f_{iI}^{int}+f_{iI}^{ext},\;\mathrm{in}\;\Omega^{s},$$


(20)
$$f_{iJ}^{FSI}=\underset{I}{\sum }f_{iI}^{FSI,s}\left(X^{s},\;t\right)\phi_{I}\left(x_{J}-x_{I}^{S}\right),{\;x}_{I}^{S}\in \Omega_{\phi J},$$


(21)
$$\rho^{f}\;\left(v_{i,t}+v_{j}v_{i,j}\right)=\sigma_{ij,j}+\rho g_{i}+f_{i}^{FSI},\;\mathrm{in}\;\Omega ,$$


(22)
$$v_{j,j}=0,\;\mathrm{in}\;\Omega ,$$


(23)
$$v_{iI}^{s}=\underset{J}{\sum }v_{iJ}\left(t\right)\phi_{J}\left(x_{J}-x_{I}^{S}\right),{\;x}_{J}\in \Omega_{\phi I}.$$



The following is an illustration of the IFEM method using a semi-explicit temporal integration:(1)Given the structure configuration **x**^s,n^, and the fluid velocity **v**^n^ at time step n,(2)Evaluate the nodal interaction forces **f**^FSI,s,n^ for solid material points, using Equation (19),(3)Distribute the material nodal force onto the fluid mesh, from **f**^FSI,s,n^ to **f**^FSI,n^ using the delta function as in Equation (20),(4)Solve for the fluid velocities **v**^n+1^ and the pressure **p**^n+1^ implicitly using Equations (21) and (22),(5)Interpolate the velocities in the fluid domain onto the material points, i.e., from **v**^n+1^ to **v**^s,n+1^ as in Equation (23), and(6)Update the positions of the structure using **u**^s,n+1^ = **v**^s,n+1^∆t and go back to step 1.

Notably, the fluid and solid are coupled explicitly despite the fluid’s complete implicit solution. If we reformat the fluid momentum equation (just discretized in time for clarity), we get
(24)$$\rho^{f}\left[\frac{v_{i}^{m+1}-v_{i}^{m}}{\Delta t}+v_{j}^{m+1}v_{i,j}^{m+1}\right]=\sigma_{ij,j}^{f,m+1}+f_{i}^{FSI,m}.$$

It is obvious that the solid equations are calculated using values from the previous time step and that the interaction force is not updated during the iteration. This force must depend on the fluid velocity at the moment in order for there to be a fully implicit coupling, and the fluid equations should be linearized to include the term *f*^FSI,*m*+1^.

## 3. The Model of RBC and CO Diffusion

### 3.1. Discrete RBC Model

RBCs take on a biconcave disc form in suspension culture, unlike white blood cells, which allows them to pass through capillaries. A typical RBC’s surface to volume ratio is substantially larger than that of a sphere generated by a tensioned membrane due to the biconcave disc shape. Furthermore, the biconcave disc form implies that the membrane cytoskeleton possesses both membrane rigidity and bending. The RBC membrane is modeled with a three-dimensional finite element formulation utilizing a Lagrangian description, as seen in Figure 1, to take into account both membrane rigidities and bending. As a result, an RBC is represented as a thin, flexible, three-dimensional framework that surrounds an incompressible fluid. Additionally, the cross-sectional profile’s x−y coordinates are described using the function shown below [16]:(25)$$\bar{y}=0.5\;{\left[1-{\bar{x}}^{2}\right]}^{1/2}\left(a_{0}+a_{1}{\bar{x}}^{2}+a_{2}{\bar{x}}^{4}\right)$$
with $a_{0}$ = 0.207, $a_{1}$ = 2.002, and $a_{2}$ = 1.123, and the non-dimensional coordinates $\bar{x}$ and $\bar{y}$ are scaled as x/5 and y/5 µm, respectively.

Finally, the material behavior of the RBC membrane is represented by a Neo-Hookean strain energy function:(26)$$W=C_{1}\left(J_{1}-3\right)+\frac{C_{2}}{2}{\left(J_{2}-1\right)}^{2}$$
utilizing the material characteristics *C*_1_ and *C*_2_, where *J*_1_ and *J*_2_ represent the functions of the deformation gradient C’s invariants as stated in [7,17]. In their models of RBC deformation, Eggleton et al. [18] and Pozrikidis [19] also employed the Neo-Hookean strain energy. It should be noted that various membrane material qualities can be simply applied to various constitutive laws. This is significant when thinking about sickle cells.

### 3.2. CO Diffusion and Absorption in Capillary Networks

We assume that CO tension gradients instantly achieve steady state and that the whole CO flux is caused by CO tension gradients driving CO diffusion into RBCs. The Laplace equation describes diffuse transport:(27)$$aD\nabla^{2}P=0$$
where *α* is the Bunsen solubility coefficient in lung tissue and plasma; *D* is the diffusion coefficient; *P* is the CO partial pressure.

The current study does not take into account the impact of red cell migration since CO does not equilibrate with capillary blood in the duration that blood spends in the capillary. Therefore, it is anticipated that the influence of blood flow on CO diffusion will be minimal. In the process of gas exchange, according to the speed of RBC flow in capillary (0.3–0.7 mm/s) and typical length of lung capillary (about 90 um), the time scale of transport of RBC in capillary during the gas exchange within one second (0.13–0.3 s) can be obtained, while gas diffusion can reach equilibrium in ms. Thus, we can assume that the static diffusion process is true.

P = Torr in the alveolar phase, 5 µm above the air–tissue interface, and P = 0 Torr at the inner membrane surface of the RBCs are the boundary conditions. We examined the full capillary length for each asymmetric distribution since RBCs might not be evenly spaced and, as a result, the capillary model might not be symmetrical. In air, tissue, and plasma, CO has distinct properties for diffusion. Table 2 provides a list of the properties used in the simulation from [20].

Once the distribution of *P* is determined, the diffusive flux of CO for each element is computed as CO flux
(28)$$\mathrm{CO}\;\mathrm{flux}\;(µm/s)=\alpha D\frac{\partial P}{\partial n}$$
where ∂P/∂n stands for P gradients calculated from a constant P surface in the normal direction. The total CO flow, equivalent to *D_M_* of each typical region, is obtained by
$$D_{M}=\sum^{\text{​}}\mathrm{flow}=\frac{\sum \mathrm{flux}\cdot \mathrm{area}}{P_{A}}$$
where *P_A_* is the mean CO partial pressure at the air–tissue interface.

With different RBCs running within a channel, we were able to calculate the overall CO flow in this manner.

The original mesh utilized in the modeling is depicted in Figure 2a (a cross-section view is shown). Alveolar air makes up the outer layer, vessel wall makes up the middle layer, and blood plasma makes up the inside layer. The blood plasma covers the RBC completely. The cross-sectional view of CO diffusion through the vessel wall and absorption by the RBC membrane is depicted in Figure 2b and Figure 3a, displaying a 3D representation of the CO diffusion flux. As depicted in Figure 3b, the plasma travels through the inner channel and carries the RBCs.

Besides, four sets of fluid meshes with progressively smaller mesh sizes are used to quantify the diffusion flux of CO from the wall to the RBC surface in order to examine the impact of mesh resolution on our simulation. With nodes above 1000, the solid mesh is found to have essentially no impact on simulation outcomes; consequently, a solid mesh with 1743 nodes and 8016 elements was employed throughout our simulations. However, it is possible to obtained the total CO flow of 1.103 µm^3^·s^−1^·torr^−1^ for a set of fluid mesh with 88,200 nodes and 81,792 elements. This value is within 2% of that provided by the finest mesh, which has 172,081 nodes and 162,000 elements. Because of its reasonable accuracy and efficiency, we decided to use this type of mesh for our simulations. Our simulation, which uses a time step of 0.001 s, runs for around a minute on a computer with a 2.0 GHz CPU.

## 4. Effect of RBC Deformation on CO Absorption Rate

Low absorption rates result in ineffective gas transfer, and therefore the relationship between absorption rate and blood flow rate/pressure is useful for therapeutic purposes. Let us start by thinking about how RBC deformation affects the rate of CO absorption. A capillary is filled with a few evenly distributed RBCs in a circular shape. The capillary flow’s inlet velocity is set at 50 m/s. We investigated how the deformation of the RBCs affects the diffusion capacity.

RBCs are deformed into a variety of asymmetric configurations under dynamic flow conditions, such as parachute-like shapes during capillary flow. Figure 4a–e depicts the dynamic deformation of an RBC from a bi-concave circular shape into a parachute shape. The distribution of CO flux over an RBC surface is depicted in Figure 4f. The magnitude of the flux is depicted by the vector’s length and color. It is evident that the flow distribution is not uniform. The flux is more concentrated at the deformed cell’s trailing tails than at its leading surface. At the folded trailing surface, the flux is quite low. Such an uneven distribution of gas flux is significantly increased when RBCs are grouped.

The time history of D_M_ when a single RBC is bent from a circular shape into a parachute is shown in Figure 5. D_M_ falls from 1.18 to 1.03 throughout this deformation phase, which can be attributed to a lower absorption rate caused by the RBCs’ folded tails.

The “breath-holding” and “stationary state” methodologies are the two main experimental methods for calculating diffusion capacity [21]. The “breath-holding” technique, where the diffusion capacity is evaluated by the CO partial pressure change during a single respiration recorded by the kymograph, was first explicitly described by Marie Krogh in 1915 [22]. For the “steady state” method, the rate of CO absorption during continuous breathing is often used to calculate the diffusion capacity. However, because our simulation is conducted at the capillary scale rather than the full human body, it is difficult to directly compare the results of our simulation to the experimental observations.

The distribution of RBCs can also affect how much CO is taken in. The CO absorption of three different RBC distributions—uniform, random, and clustered—was investigated by Connie et al. The maximum CO uptake was observed to result from uniform distribution, whereas the worst CO uptake resulted from clustered distribution. While the actual CO uptake is carried out by 3D deformable RBCs in a dynamic process, the calculations in20 are based on circular, rigid 2D RBCs in a static condition.

We looked at the uniform, random, and cluster distributions of RBCs. Figure 6 shows the RBC position and CO flow pattern for these three RBC distributions. In Figure 7, the total diffusion capacity DM for various RBC distributions is displayed. The RBCs with a homogeneous distribution have the highest overall diffusion capability. As the RBC distribution becomes random, it declines until it is at its lowest for clustered RBCs. The total diffusion capacity rises as the number of RBCs in the capillary does. It should be noted that when the capillary is entirely occupied (in our case, the capillary is totally occupied by 7 RBCs), the overall diffusion capabilities of the three distributions equal each other out. However, as illustrated in Figure 8, the diffusion capacity per RBC diminishes as the capillary’s RBC density (blood hematocrit) rises. This is due to the fact that tightly packed RBCs reduce the CO concentration in the area, which lowers the rate of absorption. It is extremely remarkable that a simple change in RBC distribution inside capillaries from uniform to nonuniform at a constant hematocrit can lower diffusive CO uptake by more than 50%.

Figure 7 displays the capillary’s overall D_M_, as well as the D_M_ for each RBC. The total DM rises as the number of RBCs does, but the DM per RBC falls. This is because the total flow reduces as the local gas partial pressure is reduced when two RBCs are very close to one another.

It is anticipated that the diffusion capabilities at various locations in the capillary network may vary because RBCs dynamically deform and form clusters.

Figure 9a shows the natural diffusion process of CO_2_ and O_2_ carried by red blood cells in the lungs of human body. (b) shows our experimental results, that is, the effect of RBCs on the amount of gas diffusion in different aggregation and deformation states, and it can be clearly concluded that the diffusion capacity of CO decreases when RBCs deform, and it can also be shown that RBCs in the aggregated state have significantly lower diffusion capacity.

## 5. Conclusions and Outlooks

Through the use of an immersed finite element platform, we have created a computer model that connects the fluid–structure interaction problem with the gas diffusion issue. To show the viability of the created coupled model, the dynamic process of deformable RBCs flowing and gas diffusion in capillaries was investigated. Our IFEM research revealed that the RBC deformation and distribution both affect the gas diffusion capacity. The CO flux across the warped RBC membrane became nonuniform when the RBCs were deformed, which lowers the diffusion capacity. The flow of RBCs across branching capillaries will be researched in the future. We can also anticipate applying this model to other areas, such as Circulating Tumor Cells. During metastasis, CTCs are produced by primary tumors and circulate in the blood vessel [23]. We may apply the study of RBCs flow process in capillaries to circulating tumor cells since these cells have properties with RBCs, such as deformability. Through this model, the flow and metastasis of CTCs in blood vessels will be further studied.

## Figures and Tables

**Figure 1 cells-11-03987-f001:**
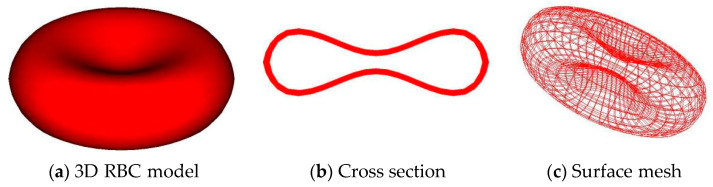
A three-dimensional finite element mesh of a single RBC model.

**Figure 2 cells-11-03987-f002:**
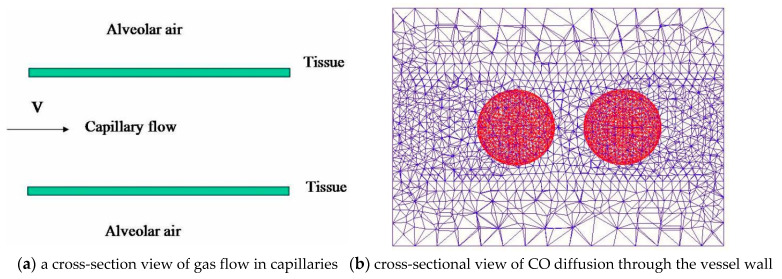
The finite element mesh of a pulmonary vessel and the uptake of CO in a pulmonary vessel.

**Figure 3 cells-11-03987-f003:**
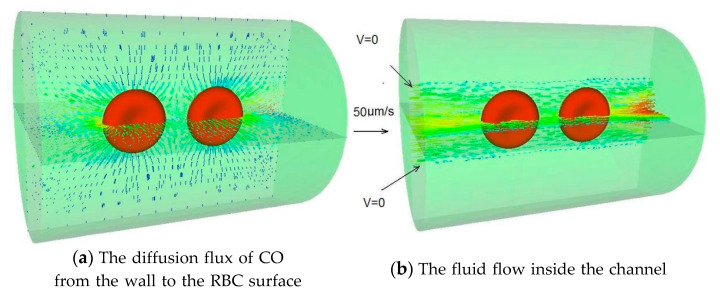
The uptake of CO in a cylindrical pulmonary vessel.

**Figure 4 cells-11-03987-f004:**
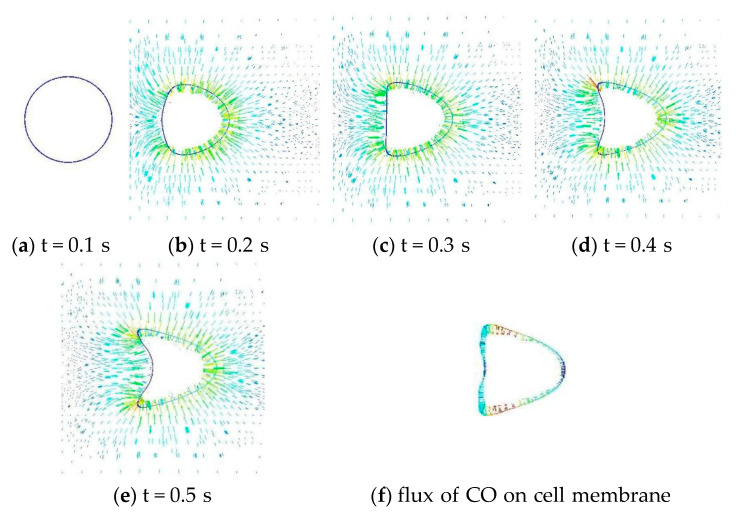
The deformation of an RBC under capillary flow in a cylindrical pulmonary vessel and the uptake of CO during this process.

**Figure 5 cells-11-03987-f005:**
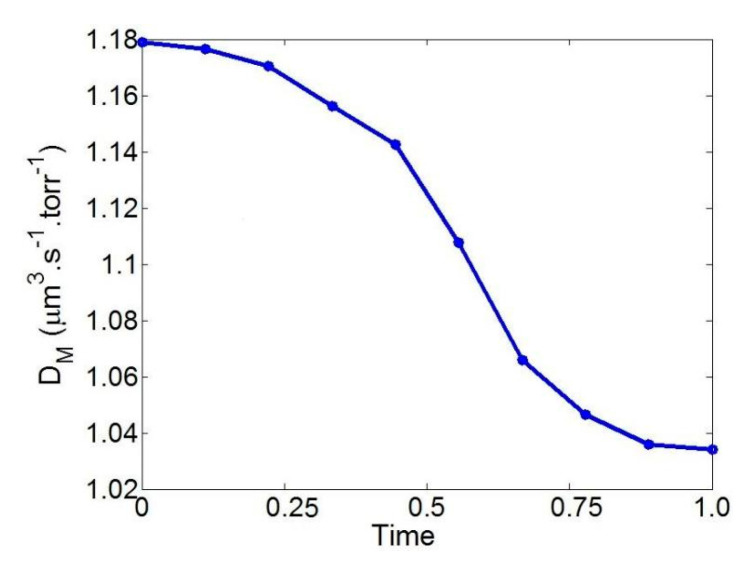
The time history of D_M_ during the deformation of a single RBC. The time is normalized with a total time of 0.8 s.

**Figure 6 cells-11-03987-f006:**
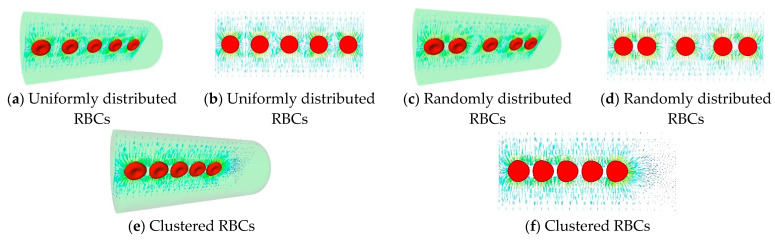
The CO flux for various RBCs distributions.

**Figure 7 cells-11-03987-f007:**
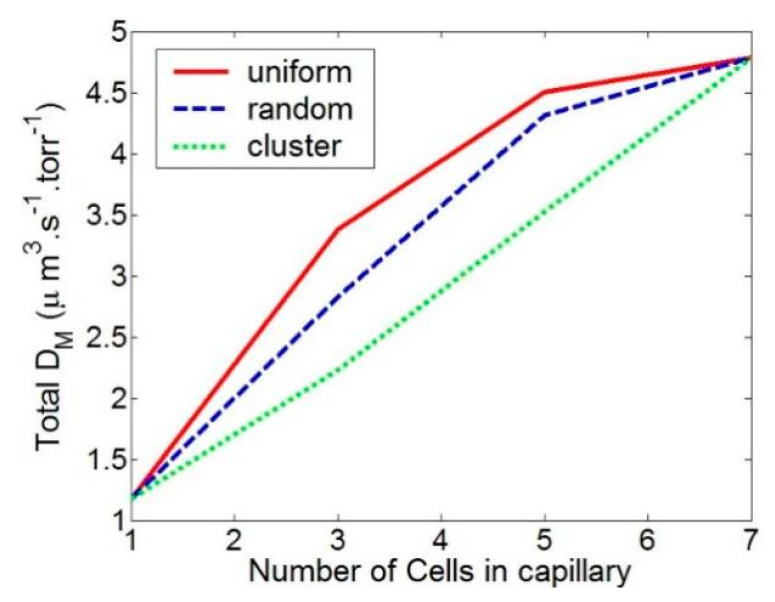
Total diffusion capacity for different RBC distribution and Hematocrit.

**Figure 8 cells-11-03987-f008:**
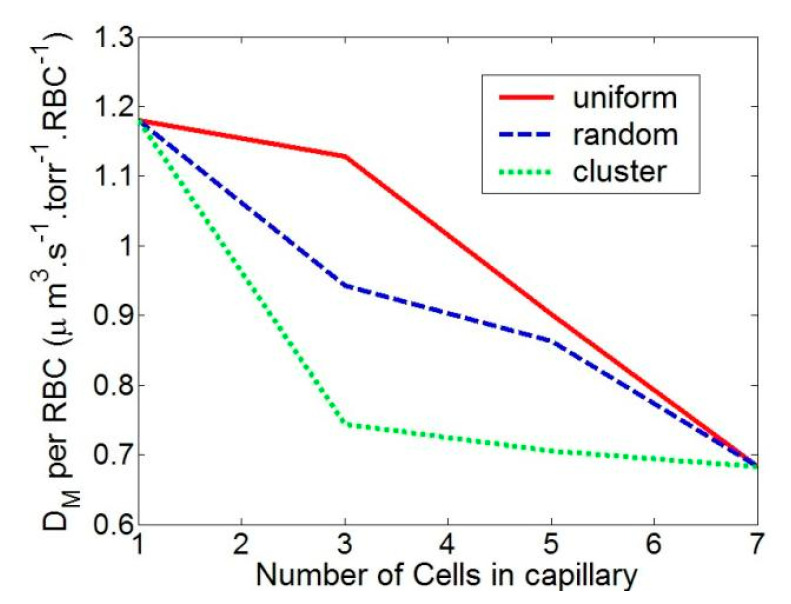
Average diffusion capacity per RBC for different RBC distribution and Hematocrit.

**Figure 9 cells-11-03987-f009:**
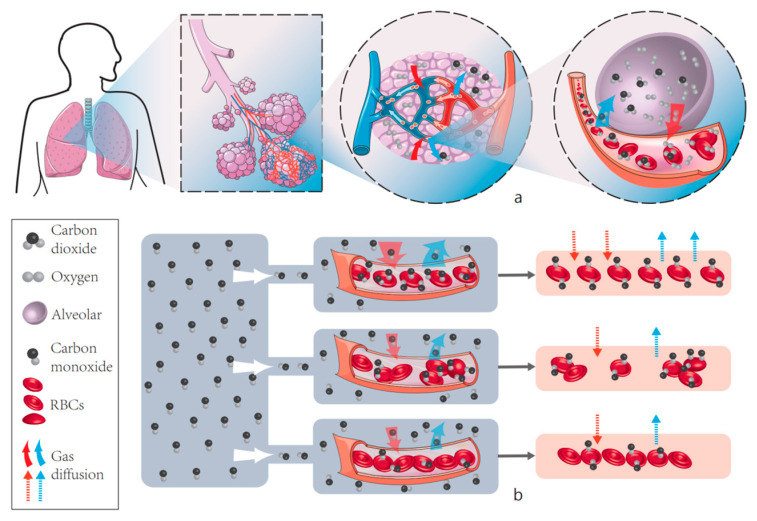
Pulmonary gas diffusion process and CO diffusion capacity of red blood cells under different aggregation states. (**a**) shows the natural diffusion process of CO_2_ and O_2_ carried by red blood cells in the lungs of human body. (**b**) shows our experimental results, that is, the effect of RBCs on the amount of gas diffusion in different aggregation and deformation states, it can be clearly concluded that the diffusion capacity of CO decreases when RBCs deform, and it can also be shown that RBCs in the aggregated state have significantly lower diffusion capacity.

**Table 1 cells-11-03987-t001:** IFEM Nomenclature.

	Fluid Domain (Ω)	Solid Domain (Ω*^s^*)
spatial coordinate	x	x*^s^*
displacement	-	u*^s^* = x*^s^* − X*^s^*
velocity	v	v^s^ = du^s^/d^t^ = ú^s^
acceleration	-	a^s^ = d2u^s^/dt^2^ = ü^s^

**Table 2 cells-11-03987-t002:** List of constants used in the simulation.

Capillary diameter	10.0 µm
RBC diameter	7.5 µm
Alveolar *P*	1.0 Torr (1 Torr = 1 mmHg)
D in air	2.413 × 10^7^ µm^2^/s
D in Tissue and plasma	2.453 × 10^3^ µm^2^/s
*α*	2.363 × 10*^−^*^5^ Torr*^−^*^1^

## Data Availability

The study did not report any data.

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
