# Peer review of "Modeling of Red Blood Cells in Capillary Flow Using Fluid–Structure Interaction and Gas Diffusion"

_cells, 2022, doi:10.3390/cells11243987_

Round 1

Reviewer 1 Report

The paper is devoted to the simulation of fluid-structure interaction between red blood cells with the influence gas diffusion. So, according to the processes of deformation in the fluid flow, the influence of gas diffusion is taken into account. For the numerical solution, the immersed finite-element method is applied. The results of simulation are presented and briefly discussed.

From my point of view, the paper is interesting and can be considered as actual for the specialists on simulation of red blood cells interaction and application of FEM for solution of complex physical problems. But the paper needed the major revision for being accepted.

Comments:

1. The more than one half of the paper is devoted to the description of the theory of well-known numerical method for modeling of fluid-structure interactions. But there no any words on the practical aspects of the used approach - on the mesh resolution, the accuracy order, stability conditions, method for the solving of algebraic systems. Additionally, the authors try to publish their manuscript in Cells, so there must be included some phrases on the advantages and disadvantages of the used approach to simulation, on the restrictions of the model, on the comparison with experimental data (if it is exist) and so on.

2. In the paper the case of the stationary diffusion is considered (the problem for eq. (22) is solved). So please include the phrases, which can justify this simplification.

Author Response

Reviewer 1: The paper is devoted to the simulation of fluid-structure interaction between red blood cells with the influence gas diffusion. So, according to the processes of deformation in the fluid flow, the influence of gas diffusion is taken into account. For the numerical solution, the immersed finite-element method is applied. The results of simulation are presented and briefly discussed.

From my point of view, the paper is interesting and can be considered as actual for the specialists on simulation of red blood cells interaction and application of FEM for solution of complex physical problems. But the paper needed the major revision for being accepted.

We thank the referee for the positive and encouraging comments.

  • The more than one half of the paper is devoted to the description of the theory of well-known numerical method for modeling of fluid-structure interactions. But there no any words on the practical aspects of the used approach - on the mesh resolution, the accuracy order, stability conditions, method for the solving of algebraic systems. Additionally, the authors try to publish their manuscript in Cells, so there must be included some phrases on the advantages and disadvantages of the used approach to simulation, on the restrictions of the model, on the comparison with experimental data (if it is exist) and so on.

We appreciate the detailed comments and constructive suggestions. To demonstrate the practical aspects of the simulation methods used, we added the following: Besides, four sets of fluid meshes with progressively smaller mesh sizes are used to quantify the diffusion flux of CO from the wall to the RBC surface in order to examine the impact of mesh resolution on our simulation. With nodes above 1000, the solid mesh is found to have essentially no impact on simulation outcomes; consequently, a solid mesh with 1743 nodes and 8016 elements is employed throughout our simulations. However, it is possible to obtained the total CO flow of 1.103µm3.s-1.torr-1  for a set of fluid mesh with 88,200 nodes and 81,792 elements. This value is within 2% of that provided by the finest mesh, which has 172,081 nodes and 162,000 elements. Because of its reasonable accuracy and efficiency, we decided to use this type of mesh for our simulations. Our simulation, which uses a time step of 0.001 seconds, runs for around a minute on a computer with a 2.0 GHz CPU.

As for the advantages and disadvantages and limitations of the simulation method proposed by the reviewer and the comparison of experimental data, Our explanation is RBCs take on a biconcave disc shape, making it better to use a 3D deformable RBC model than a 2D rigid RBC model to depict the membrane rigidity and bending of the RBCs. Additionally, although the calculations are based on circular rigid 2D RBCs under static conditions, the actual CO uptake is performed by dynamic processes with 3D deformable RBCs. And because the small domain size and number of RBCs used in the simulation, as well as other intricate biochemical elements present in real blood that are not taken into account by our cell-cell interaction model, make it difficult to make precise comparisons to the experimental results. This issue will be addressed in our future works. However, to our knowledge, this is the first research that links the microscopic mechanism of RBC aggregation to the macroscopic blood viscosity via direct 3D numerical simulation.

  • In the paper the case of the stationary diffusion is considered (the problem for eq. (22) is solved). So please include the phrases, which can justify this simplification.

We thank the reviewer for this valuable suggestion. To demonstrate the simplified process for the stationary diffusion case, we add the following description: The current study does not take into account the impact of red cell migration since CO does not equilibrate with capillary blood in the duration that blood spends in the capillary. Therefore, it is anticipated that the influence of blood flow on CO diffusion will be minimal. In the process of gas exchange, according to the speed of RBC flow in capillary (0.3-0.7mm/s) and typical length of lung capillary (about 90um), the time scale of transport of RBC in capillary during gas exchange is within one second(0.13-0.3s)can be obtained, while gas diffusion can reach equilibrium in ms, Thus, we can assume that the static diffusion process is true.

Reviewer 2 Report

This is a paper on new method to study the carbon dioxide (CO2) diffusion according to the shape of the red blood cell (RBC). 

1. The authors mentioned the diffusion of carbon monoxide (CO) throughout the text.  This is a major mistake. In normal patient, the main job of the RBC is to release the CO2 to the alveoli and not CO. CO is only a minute portion of the air in the lungs, except in chronic smokers or in toxic case. In normal pulmonary physiology, we dont talk about CO diffusion, we talk about CO2 diffusion.  

2. The authors mixed together the absorption of CO2 by the RBCs and the diffusion of CO2 out of the RBCs in the lungs.  

These are 2 different processes: absorption in the capillaries or veins of the organs and diffusion out of the RBCs in the capillaries of the lungs.   

Besides this problem, the methodology is GREAT. The model is great too. The conclusion is that the authors create a new model to study the change of CO2 diffusion based on the shape of the RBCs when going through the capillaries.   

Author Response

Reviewer 2: This is a paper on new method to study the carbon dioxide (CO2) diffusion according to the shape of the red blood cell (RBC). 

We appreciate the detailed comments and constructive suggestions.

  • The authors mentioned the diffusion of carbon monoxide (CO) throughout the text.  This is a major mistake. In normal patient, the main job of the RBC is to release the CO2 to the alveoli and not CO. CO is only a minute portion of the air in the lungs, except in chronic smokers or in toxic case. In normal pulmonary physiology, we don’t talk about CO diffusion, we talk about CO2
  • The authors mixed together the absorption of CO2 by the RBCs and the diffusion of CO2 out of the RBCs in the lungs.  

These are 2 different processes: absorption in the capillaries or veins of the organs and diffusion out of the RBCs in the capillaries of the lungs.   

Besides this problem, the methodology is GREAT. The model is great too. The conclusion is that the authors create a new model to study the change of CO2 diffusion based on the shape of the RBCs when going through the capillaries. 

We would like to express our sincere thank to the reviewer for the constructive comments. However, the investigations on the diffusing capacity of the lungs for carbon monoxide (DLCO, also known as transfer factor for carbon monoxide or TLCO) are significant for clinical purpose. Nowadays, the test of DLCO is one of the most clinically valuable tests of lung function. Actually, the earliest investigation on the DLCO was conducted in 1909 by Marie Krogh (Scand Arch Physiol, 23 : 236-247, 1909). Up to now, the DLCO has become one of the most useful routine tests of pulmonary function in laboratories worldwide. The standards for DLCO instruments, performance of the test, and calculation of the results were initially published by the American Thoracic Society in 1987, and updated by the American Thoracic Society and European Respiratory Society in 2005 and in 2017 (American Review of Respiratory Diseases, 136: 1299, 1987; European Respiratory Journal, 26(4): 720-735, 2005; European Respiratory Journal, 49(1): 1600016, 2017). A recently published overview reveals the evolution of the measurement and estimation of DLCO in details (Comprehensive Physiology, 10: 73-97, 2020). Overall, the DLCO can be estimated from relatively simple measurements (i.e., measurements of inhaled gas volumetric-rate and concentration, and of exhaled gas concentration) and used to indicate abnormalities in the respiratory diffusion process. However, it is difficult to infer from variations in the lung diffusing capacity the precise cause of the abnormality because the sensitivity of the lung diffusing capacity (a global parameter) to local changes in the morphology and functionality of the lung is not well known. There is a need for linking the local diffusion process occurring within the alveolar region to the measurable variations in the global lung diffusing capacity parameter. Thereby, a number of models that simulate locally the gas diffusion process inside the alveolar region of the lung have been proposed. For instance, Connie C. W. Hsia and coworkers computed the uptake of CO based on a two-dimensional geometric capillary model containing a variable number of red blood cells (RBCs) in order to determine the effects of red cell distribution on pulmonary diffusive CO transport. In contrast, in this paper we estimated the DLCO based on a more realistic three-dimensional deform-able red blood cell (RBC) model. We hope that the simulation can advance the understanding of the gas, particularly, CO diffusion within the alveolar region of the lung.

Besides, we have made some explanation, explaining the absorption of carbon dioxide by red blood cells and the diffusion of carbon dioxide by red blood cells in the lungs : The process of absorption of carbon dioxide by red blood cells is that carbon dioxide passes through the tissue cell wall and then passes through the capillary wall closest to the cell, binding to hemoglobin. The diffusion process of red blood cells to carbon dioxide in the lung is that when the pressure of CO2 in the lung capillary is very low, the CO2 in the red blood cells will be released, and when the pressure of CO2 in the body capillary is high, CO2 will enter the red blood cells.

Reviewer 3 Report

In this manuscript, An et al. modeled red blood cells in capillary flow with the use of fluid-structure-interaction and gas diffusion. Overall, this is a well-written, well-organized manuscript. The method was well explained and the results were well described. I therefore recommend it for publication in Cells.

Author Response

Reviewer 3: In this manuscript, An et al. modeled red blood cells in capillary flow with the use of fluid-structure-interaction and gas diffusion. Overall, this is a well-written, well-organized manuscript. The method was well explained and the results were well described. I therefore recommend it for publication in Cells.

We thank the reviewer for evaluating our paper.

We sincerely hope that this revised manuscript has addressed all your comments and suggestions. We appreciated for reviewers' warm work earnestly, and hope that the correction will meet with approval. Once again, thank you very much for your comments and suggestions.

Round 2

Reviewer 1 Report

It is fine now!

Reviewer 2 Report

I agree with the explanations of the authors